# Peripartum Pubic Symphysis Diastasis—Practical Guidelines

**DOI:** 10.3390/jcm10112443

**Published:** 2021-05-31

**Authors:** Artur Stolarczyk, Piotr Stępiński, Łukasz Sasinowski, Tomasz Czarnocki, Michał Dębiński, Bartosz Maciąg

**Affiliations:** Department of Orthopedics and Rehabilitation, Medical University of Warsaw, 02-091 Warsaw, Poland; artur.stolarczyk@wum.edu.pl (A.S.); lukasz.sasinowski@wum.edu.pl (Ł.S.); tomasz.czarnocki@wum.edu.pl (T.C.); ortopedia@mssw.pl (M.D.); bartosz.maciag94@gmail.com (B.M.)

**Keywords:** pubic symphysis separation, pubic symphysis diastasis, pubic symphysis, pregnancy, PSD

## Abstract

Optimal development of a fetus is made possible due to a lot of adaptive changes in the woman’s body. Some of the most important modifications occur in the musculoskeletal system. At the time of childbirth, natural widening of the pubic symphysis and the sacroiliac joints occur. Those changes are often reversible after childbirth. Peripartum pubic symphysis separation is a relatively rare disease and there is no homogeneous approach to treatment. The paper presents the current standards of diagnosis and treatment of pubic diastasis based on orthopedic and gynecological indications.

## 1. Introduction

The proper development of a fetus is made possible due to numerous adaptive changes in women’s bodies, including such complicated systems as: endocrine, nervous and musculoskeletal. With regard to the latter, those changes can be observed particularly in osteoarticular and musculo-ligamento-fascial structures. Almost all of those changes have an aim to broaden space inside the pelvic ring, especially to increase the transverse diameter to provide the best conditions for fetal development and safe delivery [1].

Weight gain of a pregnant woman and a shift in the center of gravity forwards causes mechanical changes mainly in the pelvic girdle and lower limb joints. There is a tendency for deepening of lordosis in the lumbar spine, forward inclination of the pelvis, and formation of flexion contractures in the hip joints [2,3]

As a result of hormonal changes occurring during pregnancy, especially under the influence of estrogens and relaxin, water is accumulated in the body and remodelling of the collagen fiber structures occurs, which in turn leads to relaxation of tendon and ligament structures. At the same time, the abdominal muscular corset (mainly the rectus abdominis muscle) is stretched, which further impairs the ability to control the balance of the body. A few weeks before delivery, the uterus and the fetus move downwards towards the pelvic inlet, further increasing the relaxation of the ligamentous structures located in the lower areas of the pelvic girdle. The process of water retention in the pregnant woman’s body leads to greater hydration of cartilage and bone tissue. This results in the softening of cartilage of intervertebral discs, pubic symphysis, and sacroiliac joints. Most noticeable changes occur in the pubic symphysis, which during pregnancy and childbirth is most prone to stretching [2,3,4].

The phenomenon of separation of the pubic symphysis often causes pain and impairs normal life activities [5]. Because of the rare occurrence of this pathology, there is a lack of therapeutic algorithms in the literature. The creation of guidelines for the diagnosis and treatment of pubic separation is aimed to make diagnosis easier and enables faster decision-making, which may result in better therapeutic outcomes.

## 2. Pubic Symphysis Diastasis—Incidence and Preliminary Characteristics

A pubic symphysis diastasis (PSD, diastasis symphysis pubis) is defined as excessive widening of the system of anatomical structures that make up the pubic symphysis (above the physiological norm of 10 mm), occurring during pregnancy or postpartum. It is total separation or instability of the symphysis without breaking the pubic bones. It is a rare disease with incidences ranging from 1/300 to 1/30,000 [1,5]. Incidence seems to grow higher with years [6]. In the current literature, many synonyms for the separation of the pubic symphysis can be found, such as: “Pubic rupture of the pelvis”, “Pubic diastasis”, or “Postpartum symphysis pubis diastasis”, which all refer to pain associated with childbirth or pelvic instability after childbirth [2].

## 3. Anatomy and Pathophysiology of Pubic Symphysis

The pelvis is a spatially closed structure formed by the pelvic bones and sacrum. The posterior part of the ring is formed by the posterior parts of the hip bones, sacroiliac joints, and sacral bone. The anterior part of the ring includes the pubic bones and the pubic symphysis. The pubic symphysis is a synchondrosis made of a fibrous cartilage disc between the two surfaces of the pubic bones covered by hyaline cartilage, which slowly decreases in thickness with age. It is strengthened by ligaments: upper pubic, lower pubic (arcuate, subpubic), posterior pubic, and anterior pubic. The posterior pubic ligament, as thin as the membrane, passes into the periosteum of the pubic bones, while the anterior pubic ligament is a thick structure containing both transverse and diagonal fibers. It also includes fibers from the aponeurosis of the abdominal muscles (rectus abdominis and oblique external muscle), gracilis muscle, and adductor longus muscle, which significantly increases effectiveness of locking the sacroiliac joints [7]. The connection of the anterior pubic ligament with the ischiocavernosus muscles and corpora cavernosa has also been described. The greatest stability of symphysis is provided by strong and thick upper pubic ligaments and lower arches. Additionally, minimal mobility is ensured by a small rotation of 1 to 3° [7].

The pubic symphysis disc is made of fibrous cartilage, in which apart from regularly arranged, thick, type I collagen fibers, chondrocytes are deployed. An additional feature of this tissue is the low content of glycosaminoglycans (2% dry weight). The arrangement of collagen fibers reflects the forces acting on the disc. Unlike the vitreous cartilage, the interpubic disc does not have perichondrium. The width of the pubic symphysis changes with age. In a newborn, it is 9–10 mm, gradually decreasing with age. A normal width of the pubic symphysis in adults is 3–6 mm, and is larger in the anterior part than the posterior [7].

Shear forces act on the joint while bending, standing, and while standing with the leg raised, known as tensile and frictional (sliding) forces, which have different values and vectors. While walking, pubic symphysis absorbs shock from the pelvic ring. Under physiological conditions, the pubic symphysis may become loose in the range of 1–2 mm (during lower limb abduction) and may also be rotated by 1°.

During pregnancy, especially in the first trimester and during childbirth, relaxation of the structures of the pubic symphysis can be observed as a result of hormone activity [2,4,8]. The perinatal dilation of the symphysis by 3–5 mm is physiological, and returns to its original size within 5 months [7]. With soft tissue relaxation comes: anterior pelvic tilt, accentuated low-back curvature, hyperextension of the upper back, forward tilting of the neck, and backward extension of the neck [8].

## 4. Etiology

Current knowledge does not allow to define a certain cause of PSD [9]. The potential factors that could contribute to occurrence of this pathology are divided into either metabolic or mechanical reasons (post-traumatic, as well as connected with degenerative changes) (Table 1) [10]. Most specialists associate the cause of this condition with the activity of relaxin, which, in combination with other hormones, causes relaxation of the ligaments within the pelvic girdle. It has been proved that the release of relaxin does not correlate with the degree of dilation of the symphysis; however, disturbed internal regulation of its secretion with simultaneous relaxation of the ligamentous apparatus can be associated with postpartum diastasis of the pubic symphysis [11,12]. The role of relaxin is not only correlated to relaxation as a possible natural antifibrotic substance, but concentrations of serum relaxin may be investigated as a potential marker of pubic symphysis diastasis [11].

Among the mechanical factors predisposing a person to the described pathology, there should be included: disproportionate labor (i.e., heavier weight of the child in relation to the relatively narrow bone pelvis of the mother), the position of the fetus in the uterus, rapid and strong contractions during labor, a second labor period which is too long, the use of labor forceps, or a medical history of trauma in the pelvic area [13]. Additional factors that may have an influence on the occurrence of PSD are: degenerative changes in sacroiliac joints, rheumatoid arthritis, pain in the lumbosacral region of the spine, hormonal disorders, and psychosocial factors. It seems that nulliparity and multiple pregnancies are also significant risk factors [14]. In recent studies, a correlation between patient age and risk of PSD was observed [6]. However, some bigger studies did not seem to confirm these observations since the age of the mother, gestational age, gender of the baby, and body weight of the baby were not risk factors of pubic symphysis diastasis [1]. It has been observed that if a woman has PGP, there is a higher probability of this disorder recurring in subsequent pregnancies; however, the degree of their intensity cannot be estimated [15]. Other factors which may contribute to PSD are epidural anasthesia, osteomalacia, chondromalacia, and previous infections of symphysis [1,14].

Interestingly, in a murine model of pregnancy, elasticity of the pelvis was investigated. Thanks to studies on genetically modified mice, it was discovered that elastin/tropoelastin, fibrillin 1, LOXL1/Loxl1, and fibulin 5 played a vital role in processes of elastogenesis and are responsible for elasticity of the pelvic girdle during labour [16]. Further evidence has shown that it is not only proteins produced by tissues that are involved in changes, but also the cellular phenotype and morphology. This discovery indicates that problems with pubic symphysis flexibility is not only connected with the extracellular matrix, but also with cells [17]. Lately, special attention was paid to monocytes and macrophages in mouse pubic symphysis remodeling. Research showed that they are involved in the relaxation of interpubic ligaments during labour, as well as in processes of repair after pregnancy [18]. Moreover, during pregnancy, proliferation in osteochondral progenitor cells is observed, especially in the osteoligamentous junction. Notably, these cells also help in recovery of the symphysis pubis when they are involved in hyaline cartilage recovery [19]. All these discoveries give new insight into complicated processes in pubic symphysis during pregnancy.

## 5. Clinical Symptoms

Characteristic symptoms of PSD are: pain localized in the area of pubic bones that gets worse when the patient tries to walk, lift something heavy, or climb stairs. Pain most often occurs in the first and subsequent days after childbirth. The condition does not always give a specific symptom. Patients can feel well and generally do not correlate minor pain with pathological changes in the pelvic girdle. Due to the fact that many cases are asymptomatic, the exact number of patients affected is difficult to assess [20].

If pubic symphysis diastasis symptoms occur, they correlate with the amount, the point of application, and the vector of forces acting on the pubic symphysis, not with the width of the separation gap. Patients are, in a majority of cases, prone to inflammation in this region [20]. Most patients complain of those symptoms within 48 h after childbirth [21]. Manifestation of all symptoms can be observed later in time during further checks in the ambulatory. There are cases in which symptoms occur even 6 months after delivery [22].

Most often, on the first postpartum day, patients complain of pain in the pubic area during attempted mobilization. An additional feature of pain is its radiation to the sacral bone and the anterior surface of the thighs, which may mimic sciatica. Intensification of this sensation occurs when changing the position of the body from lying to standing and while lifting or climbing stairs. Standing on one leg is impossible or accompanied with severe pain [2,10,20,22]. Less commonly, swaying (duckling) gait and dysuria are observed. The most common symphysis symptoms are presented in Table 2.

## 6. Differentiation

To describe pain in the area of pelvis, the term “pelvic girdle pain” (PGP) is often used, which is defined as discomfort or pain in the two pelvic connections: the sacroiliac joints and the pubic symphysis. Most of the patients diagnosed with PGP are women undergoing pregnancy or just after its termination. Currently, it is assumed that the frequency of this disorder reaches even 45% of pregnant women and 25% in the first months of puerperium [23]. About 25% of patients experience severe pain, and in 8% of patients it causes a significant reduction in daily functioning. The percentage of patients with severe ailments during the puerperium remains at the level of 7% [23,24]. Recent analyses stress the fact that PSD is not associated with the pain severity of PGP [25].

Differentiation of pelvic pain after delivery is based on a detailed medical history, as well as physical examination. In order to clarify the clinical examination, it is recommended to ask the patient the following questions [22]:

One or more positive responses:Are you having difficulty rolling over in bed?Does pain increase/decrease when climbing stairs?Are you feeling uncomfortable during a full-length stride?Do you have difficulties getting up from low chairs?

Two or more positive responses:Does pelvic pain occur when moving small weights?Is there any pain when you roll over in bed?Is there any pain when getting up from a chair or walking upstairs?

According to Wellock and Fry et al., Diagnosis for symphysis diastasis should be performed in the case of [22,26]:-History of pelvic pain after childbirth;-After excluding other diseases;-With obvious clinical symptoms.

Differential diagnosis must exclude symptoms of sciatica, infection of the genitourinary system, deep vein thrombosis, and inflammation of the pubic symphysis [27,28,29].

During physical examination, the first change that can be spotted is the Destot sign (hematoma in labia majora) which indicates pelvic ligament injury with pelvic floor disruption [30]. Another sign is in the supine position, when a patient’s legs will involuntarily move apart [31].

The disease that most commonly mimics PSD is osteomyelitis or septic arthritis of pubic symphysis. Although clinical findings, such as pain in pubic area, problems during ambulation, and pain with hip movement are the same, there are a few differences. First, the leucocyte level, C-reactive protein level, and erythrocyte sediment rate are significantly increased compared to only mild leucocytosis in PSD. Another important finding may be seen in imaging studies, such as: abscess formation in ultrasonography and sclerosis combined with osteolytic changes in X-rays [32]. The most common pathogens involved in osteomyelitis are S. aureus and P. aeruginosa, but before results of a culture, Serratia marcescens, Streptococcus group G, Cutibacterium acnes, Staphylococcus capitis, Staphylococcus epidermidis, and Enterococcus faecalis should also be taken into consideration as potential causes [33].

Another diagnosis which should be excluded is osteitis pubis. It causes loss of flexibility in the groin region, dull aching pain in the groin, and tenderness in pubic area, which can emulate PSD. Even though it usually affects athletes, it can also occur spontaneously after childbirth. CT and ultrasonography may be helpful in differentiating this condition [34].

For proper diagnosis, sacral insufficiency fractures should also be taken into consideration. In those cases, pain is located in the area of the buttock and cause similar biomechanical changes. The following conditions predispose people to these fractures: prolonged corticosteroid treatment, rheumatoid arthritis, fibrous dysplasia, Paget disease, osteogenesis imperfecta, osteomalacia, and hyperparathyroidism. The ideal diagnostic tool for differentiation is an MRI, which shows bone marrow oedemas and even the thinnest fracture lines [35].

Less common causes of pelvic girdle pain and widening of symphysis may be neoplastic destruction of the pubis, which is typically due to multiple myeloma, metastatic adenocarcinoma, or direct infiltration from adjacent rectal or genitourinary cancers. Although they are not connected with pregnancy, those neoplasms can cause the first symptoms right after delivery which could possibly mimic PSD and lead to misdiagnoses. The most important method for differentiation in this case is an MRI [36].

There is also low back pain (LBP), which causes very similar symptoms. Pain is usually located between the posterior iliac crest and the gluteal fold near one or both sacroiliac joints, occasionally radiating into the posterior thigh. It can occur in conjunction with or separately from the pubic symphysis pain, with possible radiation into the anterior part of the thigh. Some practitioners recommend acupuncture and exercises with a rigid belt to improve average pain. However, pharmacologic management is the gold standard [8]

In case of hemodynamic instability and signs of acute PSD, immediate usage of a pelvic binder or properly placed sheet is recommended [31].

## 7. Imaging

Imaging of symphysis can include both non-invasive tests (USG, MRI), as well as those with the use of ionizing radiation (X-ray, CT). Ultrasound and magnetic resonance imaging should form the diagnostics of choice during pregnancy to avoid radiation exposure [37], and X-ray and computed tomography only after childbirth. A simple and quick way to determine the width of a symphysis pubis is the use of conventional radiological diagnostics in the form of an X-ray image in AP projections and aimed at the pelvic inlet and outlet. In the projections mentioned, it is also possible to evaluate the divergence of the sacroiliac joints [6]. Physiologically, the width of the pubic symphysis during pregnancy may reach 9 mm. A diagnosis can be made when the intrapubic gap is greater than 10 mm at the narrowest point.

To illustrate instability of the pubic symphysis, a functional test could be performed, during which the patient assumes the “flamingo” position [9]. It consists in putting a load on one limb, while the other one makes simultaneous flexion of the knee and hip joints. Vertical displacement of the pubic bones in relation to each other by 1 cm indicates instability of the juncture, and values above 2 cm indicate an additional component of instability of the sacroiliac joints [38].

The use of ultrasounds in assessment of the pubic symphysis and precise measurement of the distance between the upper branches of the pubic bones should be the gold standard. This non-invasive, fast, and easily available test can be performed in every stage of pregnancy and childbirth. The small thickness of the soft tissues and short distance between the ultrasound head and the symphysis guarantees good image quality and enables precise measurements [37]. Advantages of these kinds of study include: non-invasiveness, safety, economy, and also the possibility of further control in the clinical setting [37,38]. In complicated cases of symphysis separation, when there is difficulty in diagnosis using conventional diagnostics (X-ray, ultrasound), computed tomography should be used (CT) and/or magnetic resonance imaging (MRI).

These studies provide detailed information on the anatomy of the pubic symphysis area, as well as the involvement of the sacral joints, which is essential for proper diagnosis of pelvic pain. MRI shows greater tissue specificity for soft tissues (especially the ligaments of the sacroiliac complex, joints, and symphysis cartilage) and enables the assessment of the subchondral layer and bone marrow edema. In addition, MRIs can be performed during pregnancy due to the lack of ionizing radiation [39].

## 8. Non-Operative Treatment Approach

Treatment of symphysis pubis is often long-lasting, and its main goal is to restore pelvic stability. In the initial stage, the therapy of choice remains non-surgical [21]. The basic method is to use a pelvic (pubic) belt [Figure 1] while simultaneously lying in bed in a lateral position, and physiotherapy of the pelvic muscles [30,40]. An important aspect is use of pharmacotherapy for pain relief, based on paracetamol or short-term administration of NSAIDs (safe in the lactation period) [20]. Additionally, the painful area around the pubis should be cooled. The intensity of pain should be measured to allow gradual mobilization (moving with crutches or with a walking frame). The therapy could also involve use of a locally administered steroid [10]. General improvement should occur within 6 weeks; however, in patients treated conservatively, pain may persist for up to 6 months [10]. It is important to closely and regularly monitor the progress of treatment [6,40,41]. There have been reports of excellent results using chiropractic management, such as: trigger point release, electrical stimulation, moist heat, sacroiliac belts, and specific stabilizing exercises. They should also be considered as possible adjustments to traditional management [42]. It is essential to avoid prolonged bed rest, as it leads to numerous complications, such as: decubitus ulcers, pneumonia, urinary tract infections, thromboembolism, musculoskeletal deconditioning, neuropathy, and joint stiffness [43]. With proper treatment, near complete closure of the pubic symphysis is usually observed on X-ray or ultrasound exams within 3 months [44].

## 9. Invasive Methods of Treatment

In the absence of results from conservative therapy for symphysis, surgery should be considered [9,30]. Indications for surgical intervention include chronic pain, failure to progress in reducing symphysis separation, and recurrence of dissolution of the pubic symphysis after removal of the pelvic belt [45]. An additional indication is instability of the sacroiliac joints. The advantage of surgical treatment is that there is a possibility of early upright standing and rehabilitation [46]. The treatment of choice is open reduction with internal fixation with the dedicated reconstruction plate [47,48]. External stabilizers are rarely used because of the risk of infection at the point of insertion of pins [49]. An urgent indication for surgical treatment is acute injuries of symphysis during childbirth and damage to the genitourinary system [50]. In the case of accompanying instability and pain in the sacroiliac joints, additional stabilization of both with cortical screws is recommended [29,41]. A tendency for earlier surgical intervention can also be observed, as it decreases the recovery time and improves the overall functional outcome [51]. Another advantage of this approach is prevention of sacroiliac arthritis, which may be developed due to instability of the pelvis, which can happen after symphysiectomy [52].

## 10. Physiotherapy in the Dissolution of the Pubic Symphysis

The goal of physical therapy should be to strengthen the deep muscles of the torso and pelvic muscles. Physiotherapy should include isometric, passive, and active exercises and should not cause unilateral overload of the pelvis. In cases of significant restrictions on the mobility of patients, it is advisable to use crutches and/or wheelchairs [6]. In those cases, walking reeducation, as well as mobilization under supervision is advisable [22]. Overall, mobilization and unloading added to conservative treatment seems to improve results, resolving symptoms within 3 months [52]. Because very few studies have included exercise programs in their conservative approaches, clear recommendations for particular exercises as part of PSD treatment are difficult to develop.

## 11. PSD and Delivery

The diagnosis of symphysis pubic diastasis is not an absolute indication for termination of pregnancy by cesarean section. Even though it is considered as a complication after vaginal delivery [53], it should be decided individually by the attending physician whether delivery by cesarean section is the right choice. Despite the lack of clear guidelines, many specialists take the dissolution of the pubic symphysis over 15 mm with accompanying changes in sacroiliac joints as an indication for a caesarean section for prevention of pelvic injuries during childbirth [54]. In addition, a lack of possibility of abduction of the lower limbs during labor and the changes within the birth canal as a consequence of diastasis may constitute contraindication to natural childbirth [9]. Another fact which should be taken into consideration is that one-third of women with PSD are at risk of diastasis during subsequent vaginal delivery [14]. In those cases, special care should be provided.

To improve decision-making, we recommend use of the algorithm shown in Figure 2.

## 12. Conclusions

Peripartum pubic symphysis diastasis, although being a rare condition, may cause serious problems during the postpartum period. Due to its complicated etiology based on improper hormone release, it is difficult to prevent. There are multiple risk factors which should be recognized before delivery. It may not only shorten diagnosis time, but should also lead to proper treatment as soon as possible. Diagnosis based on examination and imaging is necessary when pelvic girdle pain occurs. There are many conditions that should be taken into account during differential diagnosis, so only a combination of different diagnostic tools can provide sufficient information. In a majority of cases, the first line of treatment is non-operative. As it is non-invasive and gives good results, it is also a preferable method. Operative treatment is suggested in cases of extensive trauma to pubis symphysis and sacroiliac joints, which may cause instability and chronic conditions leading to arthritis. In order to make decisions easier, a special algorithm is recommended to use in everyday clinical practice.

## Figures and Tables

**Figure 1 jcm-10-02443-f001:**
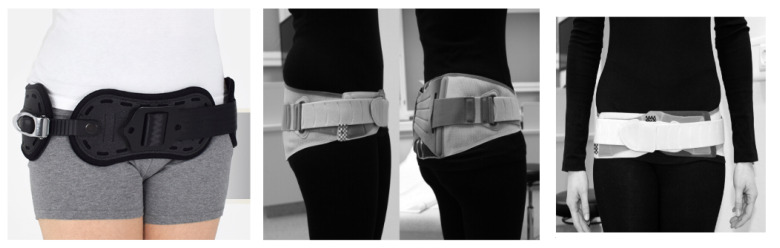
Proper positioning of the pelvic belt.

**Figure 2 jcm-10-02443-f002:**
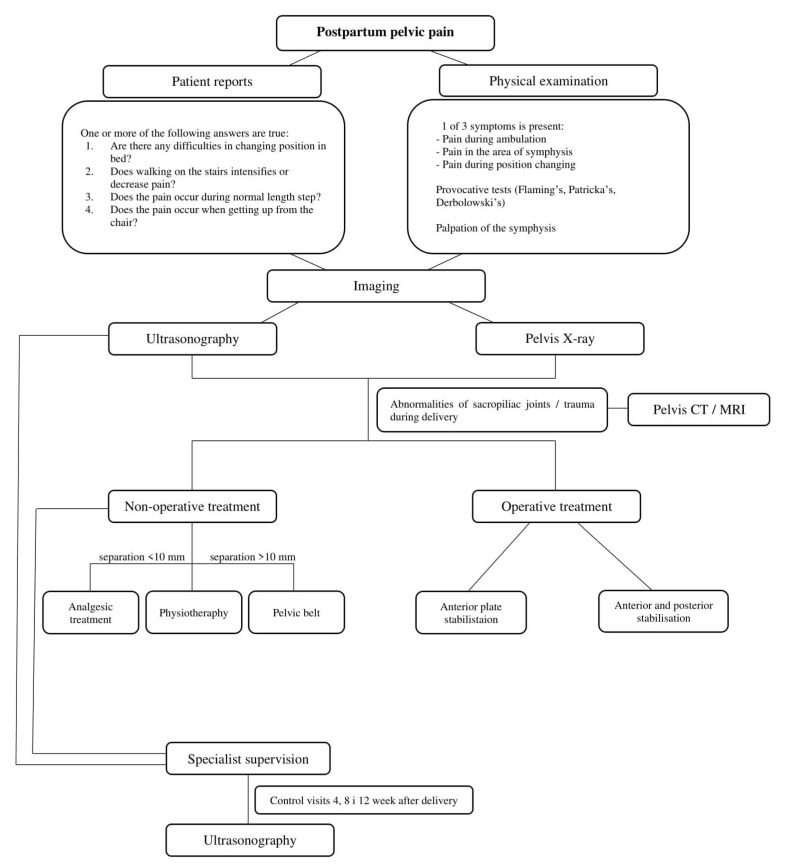
Algorithm for decision-making in symphysis pubic separation.

**Table 1 jcm-10-02443-t001:** Etiological Factors of Peripartum Pubic Symphysis Separation.

Etiological Factors of Peripartum Pubic Symphysis Separation
Congenital pelvic asymmetry, hyperlordosis, pathological pelvic dimensions Improper regulation of collagen synthesis, generalized joint laxity Increased release of relaxin, estrogen and progesterone Metabolic abnormalities of wit. D and calcium turnover Past history of pelvic trauma Inflammation of sacropelvic joints/pubic symphysis Osteoarthritic changes Macrosomia, high mother’s age, previous complications during delivery Forceps delivery Sports: football, basketball, light athletics

**Table 2 jcm-10-02443-t002:** Most Common Signs and Symptoms of Pubic Symphysis Separation.

Most Common Signs and Symptoms of Pubic Symphysis Separation
Pain radiating to sub-abdominal region, sacroiliac area, inguinal area, and lateral part of the thigh Problems with daily living activities (bending, standing on one leg, rising up, walking on the stairs, changing position in bed) Pain that wears off after rest Clicking in the area of pubic symphysis Swaying while walking (duckling) Urinary retention Urinary/fecal incontinence

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
