# Peer review of "Peripartum Pubic Symphysis Diastasis—Practical Guidelines"

_jcm, 2021, doi:10.3390/jcm10112443_

Round 1
Reviewer 1 Report
The authors discuss an infrequent but important complication of labor and postpartumThe sense of the manuscript is okay, but there are some major points to review.
1) Absolutely delete sections 6 and 8, they are short compared to the other sections and totally useless.
2) The literature is not up to date, most of the articles cited are from the 80-90s, several more recent references have been ignored that should be included and discussed. Here are some more recent references (the list is not exhaustive):
Norvilaite, Kristina, et al. "Postpartum pubic symphysis diastasis-conservative and surgical treatment methods, incidence of complications: Two case reports and a review of the literature." World journal of clinical cases 8.1 (2020): 110.
Sung, JH., Kang, M., Lim, SJ. et al. A case–control study of clinical characteristics and risk factors of symptomatic postpartum pubic symphysis diastasis. Sci Rep 11, 3289 (2021). https://doi.org/10.1038/s41598-021-82835-8
Tripathy, S. K., Samanta, S. K., Paulson Varghese, S. N. N., & Agrawal, K. (2020). Late-Onset Sacroiliac Osteoarthritis After Surgical Symphysiotomy: A Case Report. Cureus, 12(11).
Chawla, Jaya Jethra, et al. "Pubic symphysis diastasis: a case series and literature review." Oman medical journal 32.6 (2017): 510.
Becker, Ines, Stephanie J. Woodley, and Mark D. Stringer. "The adult human pubic symphysis: a systematic review." Journal of anatomy 217.5 (2010): 475-487.
Casagrande, Danielle, et al. "Low back pain and pelvic girdle pain in pregnancy." JAAOS-Journal of the American Academy of Orthopaedic Surgeons 23.9 (2015): 539-549.
Khorashadi, Leila, Jonelle M. Petscavage, and Michael L. Richardson. "Postpartum symphysis pubis diastasis." Radiology case reports 6.3 (2011): 542.
Temme, Kate E., and Jason Pan. "Musculoskeletal approach to pelvic pain." Physical Medicine and Rehabilitation Clinics 28.3 (2017): 517-537.
….. and more.
3) It might be interesting in section 3 to add some recent data on preclinical experimental models, consider (e.g.):
Consonni, Sílvio Roberto, et al. "Elastic fiber assembly in the adult mouse pubic symphysis during pregnancy and postpartum." Biology of reproduction 86.5 (2012): 151-1.
Consonni, S. R., Giardini Rosa, R., Cavinato Nascimento, M. A., Mendes Vinagre, C., Szymanski Toledo, O. M., & Pinto Joazeiro, P. (2012). Recovery of the pubic symphysis on primiparous young and multiparous senescent mice at postpartum.
Castelucci, B. G., Consonni, S. R., Rosa, V. S., & Joazeiro, P. P. (2019). Recruitment of monocytes and mature macrophages in mouse pubic symphysis relaxation during pregnancy and postpartum recovery. Biology of reproduction, 101(2), 466-477.
Castelucci, Bianca Gazieri, et al. "Time-dependent regulation of morphological changes and cartilage differentiation markers in the mouse pubic symphysis during pregnancy and postpartum recovery." PLoS One 13.4 (2018): e0195304.
4) I recommend an English revision. For example, there are a lot of very short sentences in the mansucript that compromise reading:
lines
34-36
37-38
46-47
59
62,63.
… and more
5) You have used the term pubic symphysis diastasis with the corresponding acornymon (PSD) only twice in the whole manuscript !!!! Please, use the acronym correctly in the text.
6) In the manuscript on several occasions you have used the term "pubic symphysis" instead PSD, e.g. 62-63: "PUBIC SYMPYSHIS is a relative indication for caesarean section [45]." It makes no sense!
7) Not enough argument is developed on the differential diagnosis with other causes of postpartum pelvic pain.
Consider these papers and develop some paragraphs on differential diagnosis:
Knoeller SM, Uhl M, Herget GW. Osteitis or osteomyelitis of the pubis? A diagnostic and therapeutic challenge: report of 9 cases and review of the literature. Acta Orthop Belg. 2006;72:541–8.
Yan CX, Vautour L, Martin MH. Postpartum sacral insufficiency fractures. Skelet Radiol. 2016;45:413–7.
Brumfield CG, Hauth JC, Andrews WW. Puerperal infection after cesarean delivery: evaluation of a standardized protocol. Am J Obstet Gynecol. 2000;182(5):1147–51.
Cosma, S., Borella, F., Carosso, A., Ingala, A., Fassio, F., Robba, T., ... & Benedetto, C. (2019). Osteomyelitis of the pubic symphysis caused by methicillin-resistant Staphylococcus aureus after vaginal delivery: a case report and literature review. BMC infectious diseases, 19(1), 1-6.
Author Response
Thank you very much for very constructive review
1) Sections were deleted
2) Literature was enriched with more up to date papers
3) Special paragraph in section 3 was added
4) Sentences were elongated, English revision was done
5) Acronym is now used more frequently with correct form (PSD)
6) Mistakes were corrected
7) Section "differential diagnosis" was enriched with new arguments
Reviewer 2 Report
The authors give an overview over a condition, pregnant women may suffer from, namely the pubic symphysis disruption, associated with pregnancy.
They try to summarize the spare literature and provide a treatment algorithm for this condition.
Though this is an interesting topic for orthopedic surgeons as well as obstetricians, it seems to me that this article is quite a "raw version". Beside the general need for revision of English language, some of the paragraphs are kind of listing of key words (e.g. paragraphs 8/9/10/11). Moreover, paragraph 6 seems to be a copied from the instructions for authors without any relation to the topic.
Author Response
Thank you very much for a review
1) Sections 6 and 8 were deleted
2) Literature was enriched with more up to date papers
3) Special paragraph in section 3 was added
4) Small mistakes were corrected
5) Section "differential diagnosis" was enriched with new arguments
6) Sections 9, 10, 11 were slightly changed to improve reading
Reviewer 3 Report
Review of article
Peripartum pubic symphysis separation – practical guidelines refers to imports clinical problems suffered quite commonly by pregnant women.
The main problem in my opinion is methodology. It is does not fulfill the criteria of review article, systematic review of meta-analysis. The article should undergo deep changes and be converted to systematic review.
Introduction is very academic and describe ie. anatomy, etiology, symptoms of pubic symphysis separation what is rather well known and can be found in books for medicine students. It does not extend our knowledge regarding this important problem. Also other sections clinical symptoms, imaging and treatment are based on commonly known practical guidelines for medical doctors and physiotherapist.
In conclusion literature citations should not be included.
Literature is based on local book and many articles from 20th century. There is no position form the last 5 years. There is a need for new and deep literature search.
Author Response
Thank you very much for a review.
1) Article underwent mild changes, even though it is still not a systematic review, deep literature search and analysis was performed
2) First parts of manuscrit were changed and enriched with more scietific reports from last years
3) Literature was revised and new deep search was performed
Round 2
Reviewer 1 Report
The manuscript has been revised and improved, but there are still problems, especially in the choice of references.
There are still many old references, and several suggested recent works have not been considered.
For example, reference 34 on osteomyelitis is from 2010; this more recent paper (https://doi.org/10.1186/s12879-019-4595-x) reviews all cases of osteomyelitis of the peri-post partum which should be included and has not been considered.
Choose the references in the text more accurately.
Cite multiple works on a single concept if necessary, but generally, choose the most recent.
Author Response
Thank you for another constructive review.
1) The oldest references were revised and deleted
2) reference with osteomielitis and a few other were added with mild changes in the text.
3) reference accuracy were slightly improved
4) number of multiple citation were changed where it was possible, most recent works were preferred
Reviewer 3 Report
In my opinion the manuscript should be converted to meta analisis.
Author Response
Thank you once again for prompt response.
There is very important point in making meta-analysis in this topic, since the knowledge should be organized well and easily accesed. So far there are no randomised studies in this field and almost all literature is based on case reports only. Since that there is a need for more studies on bigger groups and then, meta analysis would be a great conclusion.